# A Pharmacological Investigation of the TMEM16A Currents in Murine Skeletal Myogenic Precursor Cells

**DOI:** 10.3390/ijms25042225

**Published:** 2024-02-13

**Authors:** Marina Sciancalepore, Asja Ragnini, Paola Zacchi, Violetta Borelli, Paola D’Andrea, Paola Lorenzon, Annalisa Bernareggi

**Affiliations:** Department of Life Sciences, University of Trieste, I-34127 Trieste, Italy; msciancalepore@units.it (M.S.); asja.ragnini@phd.units.it (A.R.); pzacchi@units.it (P.Z.); borelliv@units.it (V.B.); dandrea@units.it (P.D.); plorenzon@units.it (P.L.)

**Keywords:** TMEM16A, Piezo1, Ani9, TMEM16inh-A01, Yoda1, skeletal muscle, myogenesis

## Abstract

TMEM16A is a Ca^2+^-activated Cl^−^ channel expressed in various species and tissues. In mammalian skeletal muscle precursors, the activity of these channels is still poorly investigated. Here, we characterized TMEM16A channels and investigated if the pharmacological activation of Piezo1 channels could modulate the TMEM16A currents in mouse myogenic precursors. Whole-cell patch-clamp recordings combined with the pharmacological agents Ani9, T16inh-A01 and Yoda1 were used to characterize TMEM16A-mediated currents and the possible modulatory effect of Piezo1 activity on TMEM16A channels. Western blot analysis was also carried out to confirm the expression of TMEM16A and Piezo1 channel proteins. We found that TMEM16A channels were functionally expressed in fusion-competent mouse myogenic precursors. The pharmacological blockage of TMEM16A inhibited myocyte fusion into myotubes. Moreover, the specific Piezo1 agonist Yoda1 positively regulated TMEM16A currents. The findings demonstrate, for the first time, a sarcolemmal TMEM16A channel activity and its involvement at the early stage of mammalian skeletal muscle differentiation. In addition, the results suggest a possible role of mechanosensitive Piezo1 channels in the modulation of TMEM16A currents.

## 1. Introduction 

Satellite cells represent a quiescent myogenic stem cell population located between the basal lamina and the sarcolemma of adult muscle fibers [1]. During skeletal muscle regeneration, part of this population proliferates as myoblasts to give rise to differentiated mononucleated fusion-competent cells named myocytes, which fuse with each other to form new myofibers. The latter step is a Ca^2+^-dependent process and requires myocytes to undergo bioelectric membrane modifications controlled by the expression and the activity of specific cell membrane ion channels [2]. 

It is well known that Ca^2+^-activated Cl^−^ channels (CaCCs) affect the cell membrane permeability to Cl^−^ by sensing the variation in the [Ca^2+^]_i_. Firstly identified as endogenous channels in *Xenopus* oocytes [3,4] and activated by intracellular Ca^2+^ increase upon fertilization, CaCCs were later reported to be broadly expressed in different species and tissues. They play a role in epithelial secretion [5], nociceptive responses [6], vascular smooth muscle [7], cardiac and neuronal membrane excitability [8], olfactory transduction [9,10] and the modulation of photoreceptor light responses [11].

The expression and function of CaCCs have not been extensively studied in mammalian skeletal muscle. The voltage-gated CIC-1 was the first plasmalemma Cl^−^ channel identified in this tissue [12]. Almost exclusively expressed in adult skeletal muscle, CIC-1 channel activity provides ~80% of the resting membrane conductance [13] and contributes to action potential repolarization. However, the CIC-1 mRNA was found to be orders of magnitude lower in myogenic precursors, suggesting that in undifferentiated skeletal muscle cells it is not translated into functional channels [14].

Among the CaCCs expressed in skeletal muscle, the role of TMEM16A channels is emerging. The TMEM16 family (TMEM16A-K) includes phospholipid scramblase and/or ion channels [15]. Among them, TMEM16A and TMEM16B are CaCCs. In particular, TMEM16A (also named Anoctamine-1, ANO 1) channels were first identified in heterologous systems by three groups independently [16,17,18] and subsequently characterized by a strong outward rectification [19].

With regard to skeletal muscle, CaCCs with “TMEM16A-like” properties were first identified in the plasmalemma of developing chick muscle cells as ion channels activated by both intracellular Ca^2+^ and depolarization and characterized by deactivating tail currents. They were proposed to be involved in long-duration spikes (LDS), membrane depolarization, myoblast fusion competence and spontaneous contractions [20]. Only after a number of years was the presence of functional TMEM16A channels demonstrated in zebrafish myotubes, where they have been proposed to be gated by ryanodine receptor 1 (RyR1)-mediated Ca^2+^ release during excitation–contraction coupling and to participate in the action potential repolarizing phase, together with delayed K^+^ currents, speeding in this way the muscle contractions for a more potent swimming [21]. More recently, in mice, the TMEM16A channels were described as cytosolic in undifferentiated myogenic precursors and present at the sarcolemma level in fully differentiated skeletal muscle fibers. In undifferentiated myogenic precursors, they were supposed to regulate IP_3_R-mediated Ca^2+^ release [22]. In addition, since in TMEM16A KO mice the lack of the channel protein was reported to stimulate proliferation and to inhibit differentiation of myogenic precursors, TMEM16A channels were proposed as important molecular players during myogenesis [22]. In contrast, the biological function of the sarcolemmal TMEM16A channels in adult mouse skeletal fibers still remains unknown. Although mostly based on immunofluorescence analysis, the observations conducted in the mouse model suggest a migration of the TMEM16A channels from the intracellular compartment to the sarcolemma and a potential switch of their cellular function during myogenesis in mammals.

Similarly to TMEM16A channels, mechanically activated Piezo1 channels play an important role in mouse skeletal muscle regeneration by controlling the activity of the myogenic precursors [23,24,25]. These channels, discovered in Patapoutian’s lab in 2010 [26], are permeable to Ca^2+^ [26,27] as well as to monovalent alkali cations (K^+^ > Cs^+^ ≅ Na^+^ > Li^3+^) [28] and can also interact with intracellular organelles, as reported in other cell types [29]. Moreover, they can be activated by the selective agonist Yoda1 [30].

In this study, for the first time, the TMEM16A currents were electrophysiologically recorded with the patch-clamp technique and characterized by using a pharmacological approach in mammalian mononucleated fusion-competent myogenic precursors (myocytes). The functional role of TMEM16A during myotube formation was also explored.

## 2. Results

### 2.1. Pharmacological Identification of TMEM16A Currents in Mouse Skeletal Myocytes

The mean resting membrane potential value of the 2−3-day-old myocytes, recorded in an NES, was −42.33 ± 1.79 mV (*n* = 9). To avoid possible contamination by K^+^- and Na^+^-mediated currents, all the electrophysiological patch-clamp recordings aimed at characterizing the TMEM16A currents were performed in a standard bath solution containing 150 mM TEA-Cl plus 2 µM TTX. The currents were recorded by holding the membrane potential at −40 mV with 1 s voltage steps ranging from −100 mV to +80 mV, in 20 mV increments, before being stepped back to −80 mV (Figure 1A, top).

In our experimental conditions, the recorded currents displayed a strong outward rectification at more positive potentials (Figure 1A, Ctrl), as already reported elsewhere and ascribed to CaCCs [19,31]. Since there are no specific agonists, to confirm the involvement of TMEM16A channel activity, the selective inhibitor Ani9 [19] was bath-applied at a concentration of 1 µM for 2–5 min. The blocker partially inhibited the amplitude of the currents in a voltage-dependent manner (Figure 1A, +Ani9). A representative example of Ani9-sensitive currents calculated by subtraction is shown in Figure 1A (Ani9-sensitive current); the characteristic outward rectification is clearly visible. The Ani9 blockage was voltage-dependent as confirmed by the comparison of the I-V relationships measured in control conditions and in the presence of the drug (Figure 1B). On average, Ani9 did not significantly affect the currents recorded at negative membrane potential (e.g., −80 mV: from −43.04 ± 18.58 pA to −36.73 ± 16.35 pA, *n* = 8, *p* > 0.05), while the reduction was detectable at positive potentials reaching significance at +80 mV (e.g., + 80 mV: from 162.30 ± 56.20 pA to 110.20 ± 39.42 pA, *n* = 8, Figure 1C).

The next set of recordings was aimed at investigating the Cl^−^ nature of the Ani9-sensitive currents calculated by subtraction in a 150 mM TEA-Cl-based standard bath solution and in 80 mM TEA-Cl (low [Cl^−^]_e_). In low [Cl^−^]_e_ conditions, the outward rectification was reduced, as confirmed by the I-V relationships of Ani9-sensitive currents calculated in Ctrl vs. low [Cl^−^]_e_ conditions (Figure 2A).

Moreover, the reversal potential of the Ani9-sensitive currents was positively shifted from −43.69 ± 9.58 mV (Ctrl) to −7.20 ± 2.42 mV (low [Cl^−^]_e_, *n* = 8, Figure 2B).

Similarly to the Ani9 effect, the aminophenylthiazole TMEM16A-selective blocker T16Ainh-A01 used at a concentration of 5 µM [19] significantly reduced the evoked Cl^−^ current amplitude (Figure 3A) but at both negative and positive potentials (−80 mV: from −67.18 ± 38.90 mV to −49.07 ± 32.60 mV; +80 mV: from 96.22 ± 38.90 pA to 69.23 ± 32.60 pA, *n* = 6). The T16inh-A01-sensitive component calculated by subtraction displayed the characteristic outward rectification of TMEM16A currents (Figure 3C,D).

### 2.2. The Blockage of TMEM16A Currents Reduces Myoblast Fusion

Thus far, Ca^2+^-driven Cl^−^ currents have been proposed to regulate myogenesis possibly through the intervention of kinase and phosphatase activity [22,32]. Since CaCCs have been proposed to be involved in myoblast fusion [20], we investigated the possible participation of TMEM16A channels in myotube formation. The effect of the specific TMEM16A antagonist Ani9 (3, 5, 10 and 30 μM) was studied in the fusion-competent myoblasts quantifying the fusion index (Figure 4). The fusion index was significantly reduced by ~37% starting from the concentration of 5 µM Ani9 (control: 18.26 ± 0.87%, *n* = 19 o.f.; 5 μM Ani9: 11.56 ± 0.60%, *n* = 16 o.f.). This effect was even more evident (~ 55%) at 10 µM Ani9 (8.33 ± 0.47%, *n* = 15 o.f., P < 0.001). At 30 µM Ani9, a significant reduction in the number of cells was observed, suggesting a toxic effect of the antagonist at the highest concentration tested in our experimental conditions (data not shown).

### 2.3. The TMEM16A Currents Are Modulated by the Piezo1 Agonist Yoda1

Mechanically activated Piezo1 channels are permeable to Ca^2+^, and their activity is of crucial importance in muscle cell precursors [24]. Western blot analysis confirmed the expression of both TMEM16A and Piezo1 channel proteins in the fusion-competent myoblasts (Figure 5A), in line with what has previously been reported in other studies [22,23].

Thus, we planned experiments to investigate the possible functional interaction between Piezo1 and TMEM16A channels.

The effect of the selective Piezo1 channel agonist Yoda1 (3 µM) on the TMEM16A chloride currents was therefore tested in the standard bath solution (Figure 5B). Interestingly, Yoda1 induced an increase in the Cl^−^ current amplitude with a strong outward rectification (Figure 5C). The effect appeared to be voltage-dependent, detectable at positive potentials and significant at +80 mV (from 76.18 ± 56.74 pA to 175 ± 65.32 pA in the presence of Yoda1, *n* = 8, Figure 5D).

To assess the specific activity of Yoda 1 on TMEM16A currents, the agonist was applied in the presence of 5 µM T16Ainh-A01. As expected, T16Ainh-A01 alone reduced the endogenous outward chloride currents but also abolished the Yoda1 effect (Figure 6A,B), suggesting a functional interaction between Piezo1 and TMEM16A currents.

The mean current amplitude recorded at +80 mV was 196.40 ± 89.47 pA in the Ctrl, reduced to 125.80 ± 53.11 pA in the presence of T16inh-A01, and was 140.2 ± 63.88 pA in Yoda1 plus the TMEM16A blocker (*n* = 5, Figure 6B).

## 3. Discussion

In this work, for the first time, the presence of TMEM16A-mediated Cl^−^ currents was demonstrated in the sarcolemma of mouse fusion-competent myocytes, suggesting an important role at the early stage of mammalian skeletal muscle differentiation.

The currents recorded and identified in this study as TMEM16A-mediated were characterized by voltage dependency, Cl^−^ permeability and sensitivity to specific TMEM16A blockers. The currents were predominantly carried by Cl^−^ as confirmed by amplitude reduction and a shift in the reversal potential towards more positive values in the case of the partial replacement of Cl^−^ in the extracellular medium. In addition to this, the observed outward rectifying features of the measured currents were consistent with the higher intracellular Ca^2+^ affinity at positive voltages, typical of the TMEM16A currents reported in other cell types [18]. Lastly, the currents were significantly reduced by the specific TMEM16A blockers Ani9 and T16inh-A01, respectively [19,33,34,35], albeit by possibly different mechanisms, as evidenced by the different voltage sensitivities of the blocking effects. Thus far, the TMEM16A channels have been reported to be expressed mostly intracellularly in undifferentiated myoblasts in mammals. Only in fully differentiated skeletal muscle cells have TMEM16A channels been described at the sarcolemma level [22]. In this work, using the patch-clamp technique, we demonstrated the presence of functional TMEM16A channels in mouse myocytes, i.e., in differentiated mononucleated fusion-competent cells. Our findings thus indicate a possible role for TMEM16A currents starting from the early phase of myogenesis. Accordingly, the fusion index of myoblasts into myotubes was impaired by the specific TMEM16A antagonist Ani9, revealing a role of Ca^2+^-dependent Cl^−^ channels in the cell fusion process. It must be pointed out that the efficacious Ani9 concentration on TMEM16A-mediated currents (1 µM) was lower compared to that affecting the fusion index (5 µM). The electrophysiological recordings were performed in a saline solution, whereas the fusion index was assessed in a culture medium. In the latter condition, interactions of the antagonist with serum proteins can occur, influencing its pharmacological effect [36]. Thus, the apparent discrepancy could be the result of different experimental conditions. Biophysical forces, as for example shear stress imposed on the extracellular matrix, are continuously exerted on membrane surfaces and are known to influence biological responses such as gene expression regulation. In skeletal muscle, the mechanical stress is known to promote satellite cell proliferation [37] and skeletal muscle regeneration [24,38,39]. We recently demonstrated the presence of the Ca^2+^-permeable mechanically activated Piezo1 channels in myogenic precursors in vitro [23,24]. The expression of both TMEM16A and Piezo1 channel proteins was demonstrated by Western blotting, as was the positive modulatory effect of the Piezo1 channel agonist Yoda1 on TMEM16A currents. Considering the specific Yoda1 selectivity [40,41], a novel functional interaction between Piezo1 and TMEM16A channels in the myotube formation seems to emerge from our observations. If this hypothesis is correct, the intracellular Ca^2+^ increase induced by the activation of Piezo1 channels could promote TMEM16A activity. This new intriguing aspect of skeletal myogenesis deserves further investigation. In addition to their role in regulating cell fusion, TMEM16A channels could control other aspects of myocyte physiology. Myocytes are known to be characterized by a resting membrane potential more depolarized [42,43] before the expression of the inward-rectifying K^+^ channels required for myocyte fusion [44]. In any cell type, an increase in Cl^−^ conductance can lead to cell depolarization or hyperpolarization depending on the [Cl^−^]_i_. Assuming a [Cl^−^]_i_ of ~55 mM as reported by direct measurements in myocytes [45] and a [Cl^−^]_e_ of ~160 mM as reported under physiological conditions [46], the reversal potential of Cl^−^ currents can be estimated at ~ −27 mV. Thus, if the myocyte resting membrane potential is ~−40 mV, as measured in this study and by other groups [44,47,48], the electrochemical gradient would suggest a depolarizing Cl^−^ efflux though TMEM16A channels. Interestingly, a Cl^−^ efflux mediated by undefined CaCCs was already suggested in chick skeletal myoblasts in a previous study [20]. Our results suggest TMEM16A as a new candidate in the control of chloride homeostasis required to regulate membrane excitability and/or enzyme activity [32]. In epithelial tissue, for instance, Cl^−^ homeostasis might even regulate molecular interactions among microdomains, proper phosphoinositide functions and signaling pathways responsible for the tissue stability [49,50]. Certainly, the characterization of the downstream molecular mechanism triggered by TMEM16A channel activity remains an open issue. Considering that a number of enzymes and lipid-interacting proteins are enriched in the TMEM16A complex [51], this aspect merits further ad hoc experiments in order to be clarified.

In conclusion, the identification of functional TMEM16A channels in myocytes represents an advance in the knowledge of the ion channels functionally expressed during skeletal myogenesis. Their contribution to the control of mouse myocyte fusion into myotubes indicates a role in the physiology of myogenic precursors in mammalian skeletal muscle. Moreover, the potential role of biomechanical forces in the control of TMEM16A currents mediated by Piezo1 channels offers new perspectives for understanding the mechano-regulation of skeletal myogenic precursors.

## 4. Materials and Methods

### 4.1. Cell Cultures

Mouse primary myoblasts derived from satellite cells of the hind leg muscles of 7-day-old male Balb/c mice were kindly provided by Prof. A. Werning’s laboratory (i28 cells, [52]. Myoblast cultures were maintained in growth medium (GM, F-10, Sigma, St. Louis, MO, USA) supplemented with 20% (*v*/*v*) fetal bovine serum (FBS, Gibco, Burlington, ON, Canada), 4 mM L-glutamine (Sigma, St. Louis, MO, USA), 100 U/mL penicillin and 100 µg/mL streptomycin (Euroclone, Milan, Italy) and subcultured by standard trypsinization every 3 days. To achieve the myocyte phenotype (differentiated mononucleated fusion-competent cells), myoblasts were plated on matrigel-coated Petri dishes at low density (1 × 10^4^ cells/mL) to avoid cell fusion, and 24 h after plating the GM was replaced with a differentiation medium (DM) consisting of Dulbecco’s modified minimal essential medium (DMEM, Dulbecco’s modified Eagle’s medium, Sigma), 2% (*v*/*v*) horse serum (Sigma) and the same concentration of L-glutamine and penicillin–streptomycin as in GM. Cell cultures were maintained at 37 °C, 5% CO_2_, and used within 2–3 days in DM.

### 4.2. Electrophysiological Recordings

Resting membrane potential was measured using a conventional whole-cell patch-clamp configuration in current clamp mode (I = 0) in a normal external solution (NES) containing (in mM) 140 NaCl, 2.8 KCl, 2 CaCl_2_, 2 MgCl_2_, 10 HEPES and 10 glucose, pH 7.3 adjusted with NaOH, and a pipette solution containing (in mM) 140 K-aspartate, 10 NaCl, 2 MgCl_2_ and 10 HEPES, pH 7.3 adjusted with KOH. Cell capacitance was measured by integrating the area underlying an uncompensated capacitive transient evoked by a voltage step of +10 mV. Mean membrane capacitance was 12.95 ± 0.20 pF (*n* = 31).

Membrane currents were recorded in voltage-clamp mode, and the solutions used were as follows: standard bath (in mM: 150 TEA-Cl, 2 CaCl_2_, 10 HEPES, pH 7.4 adjusted with CsOH); pipette solution (in mM: 140 K-aspartate, 10 NaCl, 2 MgCl_2_, 10 HEPES, pH 7.4 adjusted with KOH). Currents were always measured at the end of 1 s voltage steps. The electrode resistance was 4–6 MΩ, and all experiments were performed at room temperature (20–25 °C).

Data were acquired by an MultiClamp 700B amplifier controlled by pClamp 11 software (Axon Instruments, Molecular Devices, San Jose, CA, USA) using a Digidata 1500B analogue-digital converter (Axon Instruments). Signals were sampled at 500 µs and low-passed filtered at 1kHz.

In all the recordings, the bath was grounded through a 2 M KCl agar bridge connected to an Ag/AgCl reference electrode. The value of the liquid junction potential (VLJP) was off-line subtracted from all command voltages [53].

### 4.3. Western Blotting

Myocytes were lysed in RIPA buffer (25 mM Tris-HCl pH = 7.6, 150 mM NaCl, 1% NP40, 0,5% sodium deoxycholate, 0,1% SDS and 1 mM EDTA) supplemented with 1× EDTA-free protease inhibitor cocktail tablets (Sigma) for 30 min at 4 °C. The insoluble debris was removed by centrifugation (13,000 g at 4 °C for 10 min) and protein concentration was determined by Bio-Rad DC Protein assay. Then, 30 µg aliquots of total protein extracts were separated by SDS-PAGE (8%, *wt*/*v*) and subject to Western blot analysis. Following electrophoresis, proteins were transferred onto a 0.22 µm nitrocellulose membrane (Amersham) in wet conditions. To minimize background signal, the membranes were incubated with 5% non-fat milk in TBS-0.1% Tween 20 (blocking buffer) for 1 h at room temperature with gentle shaking. Membranes were then incubated overnight at 4 °C with rabbit polyclonal antibody against Piezo1 and TMEM16A (NBP1-78446 Novus Biologicals and BS-3794R Thermo Fischer, 1:1000 and 1:500, respectively) in the blocking buffer. Primary antibodies were revealed by HRP-conjugated secondary antibodies (Sigma) followed by enhanced chemiluminescence WesternBright ECL (Advasta, San Jose, CA, USA).

### 4.4. Fusion Index

The myocyte fusion index was established by dividing the number of nuclei in the myotubes (i.e., cells with more than two nuclei) by the total number of nuclei in 3-day cultures in DM. The number of nuclei was evaluated using fluorescent staining with 4,6-diamino-2-phenylindole (DAPI). Briefly, cells grown on glass coverslips were fixed in freshly prepared 3.7% (*w*/*v*) paraformaldehyde in phosphate-buffer saline (PBS, 15 min, 22 °C). After an extensive and gentle wash-out with PBS (3 × 10 min, 22 °C), myocytes were permeabilized by an incubation in PBS combined with Triton-X 100 0.1% and 5% normal goat serum (Blocking Solution, 10 min, 4 °C). Staining was then carried out by incubating the cells in Blocking Solution containing 10 μm DAPI (60 min, 4 °C). Samples were visualized under a Leica DMLS fluorescence microscope (Leica Microsystems, Wetzlar, Germany) equipped with the software LASV4.13. The fluorescence images were captured with a 20× objective and the acquired fluorescence images were compared to the corresponding bright-field images. Nuclei count, image sizing, cropping and overlay were performed with Fiji software [54]. The myocyte fusion index was calculated in the control condition and in the presence of Ani9 (3, 5 and 10 μM) in DM in two independent cell cultures. For each experimental condition, at least 15 randomly selected optical fields (o.f.) were analyzed. In each o.f., at least 300 nuclei were counted.

### 4.5. Statistical Analysis

Statistical analysis was performed with GraphPad Prism 8 and OriginPro software. The results are shown as the mean ± standard error (SEM) values. The Shapiro–Wilk test was used for normality testing. The ratio *t*-test was utilized to compare two groups of paired and normally distributed values. Depending on whether the values were paired or not, the Wilcoxon or the Mann–Whitney test was used to compare non-normally distributed values, respectively. Data were considered statistically significant when the *p* value was <0.05 (*).

## Figures and Tables

**Figure 1 ijms-25-02225-f001:**
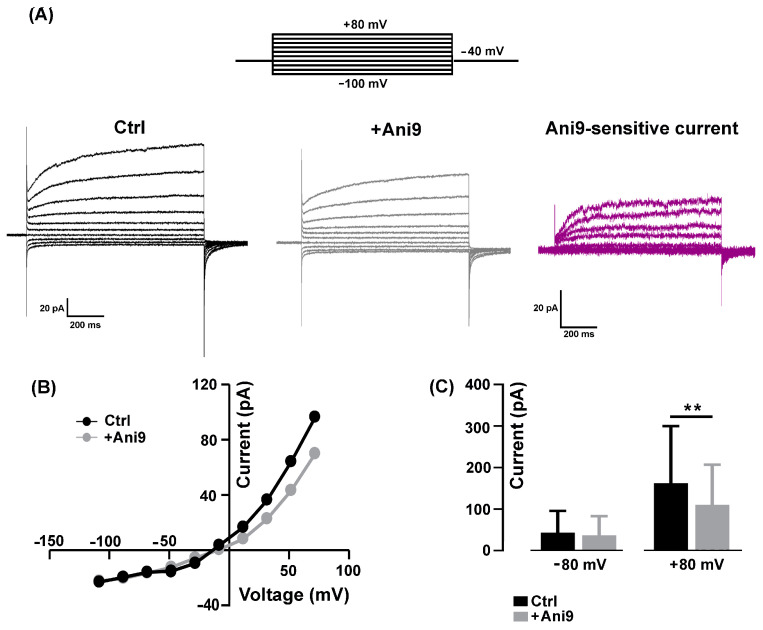
Block of TMEM16A chloride currents by Ani9. (**A**) Representative currents elicited by 1000 ms test potentials between −100 and +80 mV in the same myocyte before (Ctrl, black) and after the addition of 1 µM Ani9 (+Ani9, gray) and the corresponding Ani9-sensitive currents (magenta) calculated by subtraction. On top, the stimulation protocol used to elicit currents. (**B**) The I-V relationships in absence and presence of Ani9. (**C**) The histogram shows the blockage exerted by Ani9 that was significant only for the outward currents at +80 mV (*n* = 8, ** *p* = 0.002).

**Figure 2 ijms-25-02225-f002:**
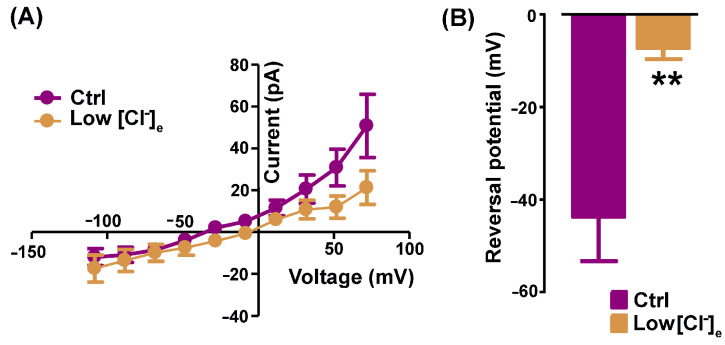
Ani9-sensitive currents are permeable to chloride. (**A**) Representative I-V relationships of Ani9-sensitive currents recorded in standard (magenta) and low [Cl^−^]_e_ bath (ochre) solutions. (**B**) The histogram shows the significant shift of the reversal potential towards a more positive value in low [Cl^−^]_e._ (*n* = 8, ** *p* = 0.006).

**Figure 3 ijms-25-02225-f003:**
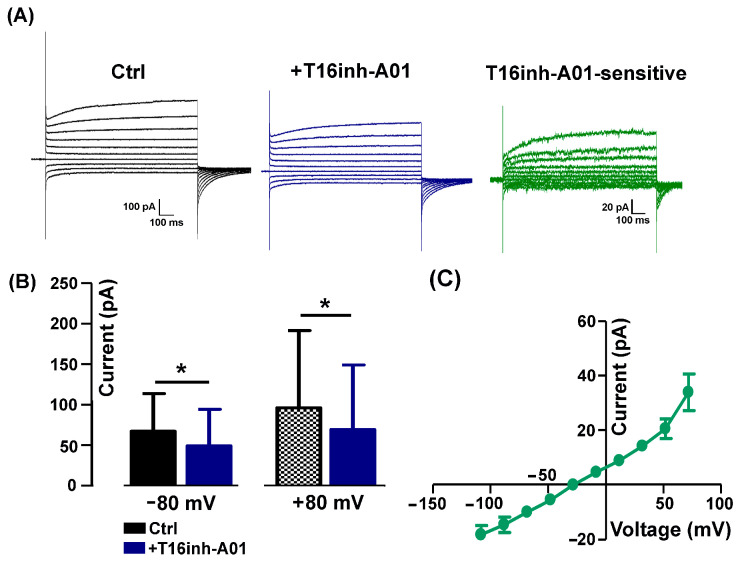
Detection of T16Ainh-A01-sensitive currents. (**A**) Representative currents recorded in absence (Ctrl, black) and in presence of 5 µM T16Ainh-A01 (+T16inh-A01, blue) and corresponding T16inh-A01-sensitive currents (green) calculated by subtraction. (**B**) In the presence of the blocker, there is a significant reduction in the currents recorded both at −80 mV and +80 mV (*n* = 6, * *p* = 0.03). (**C**) I-V relationship of T16inh-A01-sensitive currents (*n* = 5).

**Figure 4 ijms-25-02225-f004:**
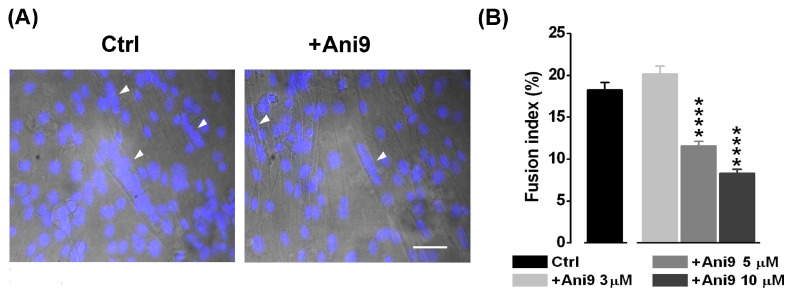
The effect of Ani9, a TMEM16A antagonist, on the myocyte fusion index. (**A**) Representative optical fields where nuclei are stained in 3-day myocyte cultures in control conditions and in the presence of 5 μM Ani9. Nuclei staining is shown as DAPI fluorescence overlay on corresponding bright-field images. Myotubes are indicated by white arrowheads. Scale bar, 50 μm. (**B**) A reduction in the mean fusion index was observed at the concentrations of 5 μM (**** *p* < 0.0001) and 10 µM Ani9 (**** *p* < 0.0001) with respect to control. Ani9 3 μM vs. 5 μM, *p* < 0.0001; Ani9 3 μM vs. 10 μM, *p* < 0.0001; Ani9 5 μM vs. 10 μM, *p* = 0.0005.

**Figure 5 ijms-25-02225-f005:**
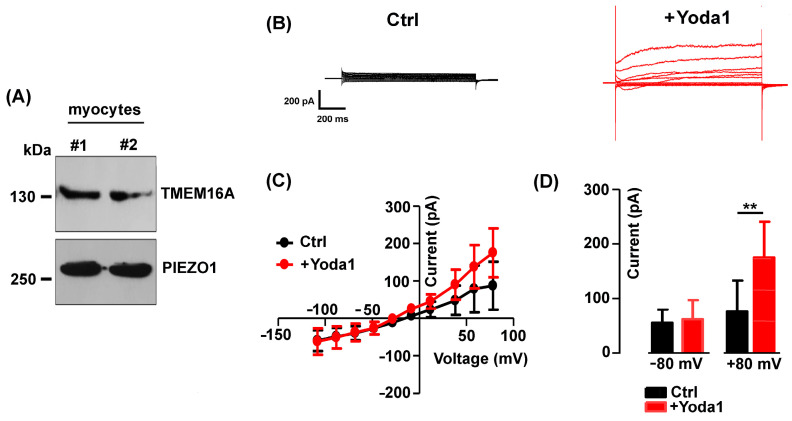
The effect of Piezo1 channel activation on TMEM16A. (**A**) Representative Western blot revealing TMEM16A and Piezo1 presence in myocyte lysate (2 preparations). (**B**) Representative currents elicited in control (black) and after 2 min in Yoda1 (3 µM, red). (**C**) Mean I-V curves before and after the addition of Yoda1 (*n* = 8). (**D**) Note the significant (** *p* = 0.008) Yoda1-induced increase in the current amplitude at positive potentials of +80 mV.

**Figure 6 ijms-25-02225-f006:**
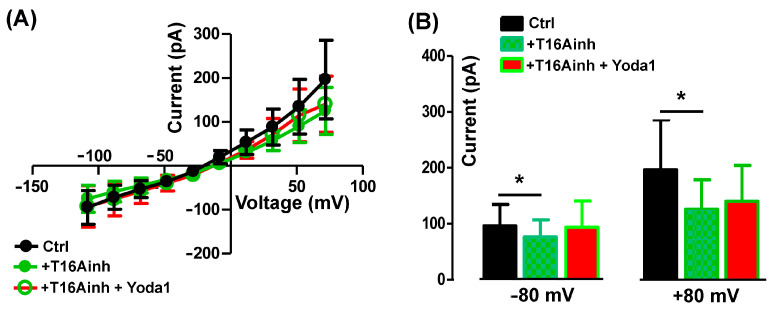
Yoda1 effect on TMEM16A currents. (**A**) I-V relationships of control (Ctrl) in presence of 5 µM T16Ainh-A01 (T16Ainh) or 5 µM Yoda 1 plus 5 µM T16Ainh-A01 (T14Ainh + Yoda1). (**B**) The TMEM16A blocker T16Ainh-A01 significantly affected the currents recorded at both −80 mV (* *p* = 0.04) and +80 mV (* *p* = 0.03). Note that the addition of Yoda1 did not induce significant changes in the presence of T16inh-A01 both at −80 mV (*p* = 0.74; *n* = 5) and +80 mV (*p* = 0.6; *n* = 5).

## Data Availability

The data presented in this study are available on request from the corresponding author.

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
