# Peer review of "A Pharmacological Investigation of the TMEM16A Currents in Murine Skeletal Myogenic Precursor Cells"

_ijms, 2024, doi:10.3390/ijms25042225_

Round 1

Reviewer 1 Report

Comments and Suggestions for Authors

The study titled "A Pharmacological Investigation of the TMEM16A Currents in Murine Skeletal Myogenic Precursor Cells" presents a comprehensive examination of TMEM16A channels in mouse myogenic precursors, shedding light on their functional expression and potential role in skeletal muscle differentiation. The combination of whole-cell patch-clamp recordings and the use of pharmacological agents such as Ani9, T16inh-A01, and Yoda1 provides a robust approach for characterizing TMEM16A-mediated currents and investigating the modulatory effects of Piezo1 channels. The inclusion of Western blot analysis to confirm the expression of TMEM16A and Piezo1 proteins further strengthens the validity of the findings.

The study's major contributions include the identification of TMEM16A channels as functionally expressed in fusion-competent mouse myogenic precursors, with their pharmacological inhibition shown to impede myocyte fusion into myotubes. Notably, the discovery of a positive regulatory effect of the Piezo1 antagonist Yoda1 on TMEM16A currents introduces a novel aspect, suggesting a potential role for mechanosensitive Piezo1 channels in modulating TMEM16A activity during the early stages of skeletal muscle differentiation. All of these findings seem valuable and suitable contribution to be published in the IJMS after justifying mentioned point .

·       How does the pharmacological blockade of TMEM16A channels impact the overall progression of myogenic differentiation, and what specific cellular events downstream of TMEM16A inhibition contribute to the observed inhibition of myocyte fusion into myotubes?

·       What molecular pathways or signaling cascades might be involved in the cross-modulation between Piezo1 and TMEM16A channels in mouse myogenic precursors?

·       Given the observed difference in Ani-9 concentration efficacy between electrophysiological recordings and the fusion index assessment, what specific factors in the experimental conditions (e.g., culture medium components, serum proteins) might contribute to this variation, and how can these conditions be optimized to reconcile the discrepancies?

·       Building on the observed positive modulatory effect of the Piezo1 channel agonist Yoda1 on TMEM16A currents, what molecular mechanisms underlie the functional interaction between Piezo1 and TMEM16A channels in the context of myotube formation? How does intracellular Ca2+ increase, induced by Piezo1 activation, influence TMEM16A activity during skeletal myogenesis?

·       Overall, this research significantly advances our understanding of TMEM16A channels in the context of skeletal muscle development and opens avenues for further exploration of their regulatory mechanisms.

Best wishes

Author Response

Review report 1.

The study titled "A Pharmacological Investigation of the TMEM16A Currents in Murine Skeletal Myogenic Precursor Cells" presents a comprehensive examination of TMEM16A channels in mouse myogenic precursors, shedding light on their functional expression and potential role in skeletal muscle differentiation. The combination of whole-cell patch-clamp recordings and the use of pharmacological agents such as Ani9, T16inh-A01, and Yoda1 provides a robust approach for characterizing TMEM16A-mediated currents and investigating the modulatory effects of Piezo1 channels. The inclusion of Western blot analysis to confirm the expression of TMEM16A and Piezo1 proteins further strengthens the validity of the findings. The study's major contributions include the identification of TMEM16A channels as functionally expressed in fusion-competent mouse myogenic precursors, with their pharmacological inhibition shown to impede myocyte fusion into myotubes. Notably, the discovery of a positive regulatory effect of the Piezo1 antagonist Yoda1 on TMEM16A currents introduces a novel aspect, suggesting a potential role for mechanosensitive Piezo1 channels in modulating TMEM16A activity during the early stages of skeletal muscle differentiation. All of these findings seem valuable and suitable contribution to be published in the IJMS after justifying mentioned point.

Overall, this research significantly advances our understanding of TMEM16A channels in the context of skeletal muscle development and opens avenues for further exploration of their regulatory mechanisms.

Authors:

We are pleased that Reviewer finds interesting our results and we would like to thank him for the positive comments

  1. How does the pharmacological blockade of TMEM16A channels impact the overall progression of myogenic differentiation, and what specific cellular events downstream of TMEM16A inhibition contribute to the observed inhibition of myocyte fusion into myotubes?

Authors:

A number of enzymes and lipid-interacting proteins are enriched in the TMEM16A complex (Perez-Cornejo P, et al., 2012). We agree with the Reviewer that the specific cellular events downstream TMEM16A activity involved in myocytes fusion into myotubes are very interesting aspects that would merit attention. However, as we already stated in the original text, ad hoc experiments would be necessary to characterize such a multifaceted molecular mechanisms. Taking into account the Reviewer comment, Discussion has been revised to further stress this point including a new reference (page 7, in red).

  1. What molecular pathways or signaling cascades might be involved in the cross-modulation between Piezo1 and TMEM16A channels in mouse myogenic precursors?

Authors:

Considering our results, it is still premature to predict the specific cross-modulation mechanisms between Piezo1 and TMEM16A channels. We think that TMEM16A could sense local variation of the intracellular Ca2+ concentration driven by the Piezo1 activation. In line with this, Piezo-mediated local Ca2+ signals might require a control of the precise distribution of Piezo1 and TMEM16A channels. Actually, the analysis of the distribution of the two channels is one of the future aims in our laboratory.

  1. Given the observed difference in Ani-9 concentration efficacy between electrophysiological recordings and the fusion index assessment, what specific factors in the experimental conditions (e.g., culture medium components, serum proteins) might contribute to this variation, and how can these conditions be optimized to reconcile the discrepancies?

Authors:

The most probable factor responsible for the discrepancy might be albumin, being the prevalent serum protein (Yang et al., 2014, now cited in the text). However, we cannot exclude other components such as serum glycoproteins and lipoproteins.

The optimization of the experimental conditions to reconcile the discrepancy of the effect of Ani9 could consist in measuring the cell fusion in serum free conditions or in a medium supplemented with a reduced serum content. However, the successful myogenic differentiation in vitro requires the presence of serum and any change in the percentage of serum during culturing would cause an impaired cell fusion per se. 

  1. Building on the observed positive modulatory effect of the Piezo1 channel agonist Yoda1 on TMEM16A currents, what molecular mechanisms underlie the functional interaction between Piezo1 and TMEM16A channels in the context of myotube formation? How does intracellular Ca2+ increase, induced by Piezo1 activation, influence TMEM16A activity during skeletal myogenesis?

Authors:

We hope to have satisfactory answered to these two questions discussing Points 1 and 2.

Reviewer 2 Report

Comments and Suggestions for Authors

The article is very well written, and presents data that in my opinion is important and a major advance in electrophysiology. These channels are very difficult to measure because the current is usually very low. However, in this work the ionic current values are high.

I would therefore like to see better explain the protocols used, as well as how the data was collected. This is because the shape of the currents is very similar to those observed with potassium currents, which are actually very high.

I'd also like to know the physiological or even pathological importance of these channels.

Author Response

Review report 2.

The authors studied the TMEM16A currents using the patch-clamp technique, moreover the functional role of TMEM16A during myotube formation was also explored. The manuscript is interesting, well written and has merit, I only have a few comments.

Authors:

We would like to thank Reviewer for the positive comments.

Since the experiment was performed on mononucleated myocytes, do authors expect that in myotubes and differentiated muscle fibres in situ TMEM16A would behave similarly.

We thank Reviewer for the stimulating question. Chloride homeostasis is crucial for function and survival of the cells. Apart its role as “inert” anion for the control of membrane excitability (especially in skeletal muscle cells), osmolarity, cell volume, and intracellular pH, it also regulates enzyme activities (Valdivieso and Santa Coloma, 2019, doi: 10.1111/brv.12536.). Considering that the electrochemical chloride gradient could be different at each stage of the myogenesis (as a result of different resting membrane potential and intracellular chloride concentrations), we expect that TMEM16A could have different functions during the myogenesis.

Please state limitations of your study.

Authors:

The first limitation is the lack of agonists for the TMEM16A channels now mentioned in the revised version (Results, subchapter 2.1, page 3, in red).

The second is that the downstream molecular mechanisms triggered by TMEM16A remain unknown. In the revised version, we stress this point in Discussion (page 7, in red).

Was expression of TMEM16A already shown in human myocytes and muscle fibres?

Authors:

The TMEM16A (Ano1) expression has been revealed by RNA sequencing in human skeletal muscle tissue and other human tissues during development (Ferrera et al., 2009, doi: 10.1074/jbc.M109.046607.). However, systematic analysis of TMEM16A expression in human myocytes vs human skeletal muscle fibres are still missing.

Reviewer 3 Report

Comments and Suggestions for Authors

The authors studied the TMEM16A currents using the patch-clamp technique, moreover the functional role of TMEM16A during myotube formation was also explored. The manuscript is interesting, well written and has merit, I only have a few comments.

1.     Since the experiment was performed on mononucleated myocytes, do authors expect that in myotubes and differentiated muscle fibres in situ TMEM16A would behave similarly?

2.     Please state limitations of your study.

3.     Was expression of TMEM16A already shown in human myocytes and muscle fibres?

Author Response

Review report 3.

The article is very well written, and presents data that in my opinion is important and a major advance in electrophysiology. These channels are very difficult to measure because the current is usually very low. However, in this work the ionic current values are high.

I would therefore like to see better explain the protocols used, as well as how the data was collected. This is because the shape of the currents is very similar to those observed with potassium currents, which are actually very high.

Authors:

We are pleased that Reviewer finds interesting our results and we are pleased to share our experience in recording the TMEM16A currents.

The TMEM16A current amplitude measured in our cell model is comparable to that recorded in other cell types such as smooth muscle cells (Bulley et al., 2012, doi: 10.1161/CIRCRESAHA.112.277145.) and endothelial cells (Ma et al., 2021, doi: 10.1016/j.jare.2020.09.003.).

We were aware of the presence of K+ currents in our cells and for this reason:

- currents were recorded in a bath solution containing the K+ blocker TEA;

- we changed the external Chloride concentration to confirm the Chloride nature of the recorded currents;

- we used specific TMEM16A blockers to obtain the TMEM16A currents by subtraction. 

I'd also like to know the physiological or even pathological importance of these channels.

Authors:

Because of their wide tissue distribution, TMEM16A channels have several physiological roles and they are supposed to be involved in many pathological conditions. The spectra of their functions are well reviewed by Bai et al., 2021 (doi: 10.1016/j.jare.2021.01.013.). At the skeletal muscle level, as stated in our manuscript, the function of TMEM16A channels has been emerging in differentiating precursors while it has not been investigated yet in adult myofibres. Therefore, in the skeletal muscle, TMEM16A roles in physiological and pathological conditions remains to be characterized.